# Predictors of mortality and loss to follow-up among drug resistant tuberculosis patients in Oromia Hospitals, Ethiopia: A retrospective follow-up study

**Demelash Woldeyohannes**[1]*, **Yohannes Tekalegn**[2], **Biniyam Sahiledengle**[2], **Tesfaye Assefa**[3], **Rameto Aman**[2], **Zeleke Hailemariam**[4], **Lillian Mwanri**[5], **Alemu Girma**[6]

1 Department of Public Health, Collage of Medicine and Health Science, Wachemo University, Hossana, Ethiopia, 2 Department of Public Health, School of Health Science, Madda Walabu University, Bale Goba, Ethiopia, 3 Department of Nursing, School of Health Science, Goba Referral Hospital, Madda Walabu University, Bale Goba, Ethiopia, 4 Department of Public Health, Collage of Medicine and Health Science, Arba Minch University, Arba Minch, Ethiopia, 5 College of Medicine and Public Health, Flinders University, Adelaide, South Australia, Australia, 6 Department of Surgery, School of Health Science, Goba Referral Hospital, Madda Walabu University, Bale Goba, Ethiopia

* woldemel@wcu.edu.et

## Abstract

### Background

Drug resistance tuberculosis (DR-TB) patients' mortality and loss to follow-up (LTF) from treatment and care is a growing worry in Ethiopia. However, little is known about predictors of mortality and LTF among drug-resistant tuberculosis patients in Oromia region, Ethiopia. The current study aimed to identify predictors of mortality and loss to follow-up among drug resistance tuberculosis patients in Oromia Hospitals, Ethiopia.

### Methods

A retrospective follow up study was carried out from 01 November 2012 to 31 December 2017 among DR-TB patients after calculating sample size using single proportion population formula. Mean, median, Frequency tables and bar charts were used to describe patients' characteristics in the cohort. The Kaplan-Meier curve was used to estimate the probability of death and LTF after the treatment was initiated. The log-rank test was used to compare time to death and time to LTF. The Cox proportional hazard model was used to determine predictors of mortality and LTF after DR-TB diagnosis. The Crude and adjusted Cox proportional hazard ratio was used to measure the strength of association whereas p-value less than 0.05 were used to declare statistically significant predictors.

### Result

A total of 406 DR-TB patients were followed for 7084 person-months observations. Among the patients, 71 (17.5%) died and 32 (7.9%) were lost to follow up (LTF). The incidence density of death and LTF in the cohort was 9.8 and 4.5 per 1000 person-months, respectively.

**Data Availability Statement:** The data underlying this study are accessible via BioStudies database

(accession number S-BSST624): http://helpdesk.
ebi.ac.uk/Ticket/Display.html?id=494263 https://
www.ebi.ac.uk/biostudies/studies/S-BSST624?
query=S-BSST624.

**Funding:** The authors received no specific funding
for this work.

**Competing interests:** The authors have declared
that no competing interests exist.

The median age of the study participants was 28 years (IQR: 27.1, 29.1). The overall cumulative survival probability of patients at the end of 24 months was 77.5% and 84.5% for the mortality and LTF, respectively. The independent predictors of death was chest radiographic findings (AHR = 0.37, 95% CI: 0.17–0.79) and HIV serostatus 2.98 (95% CI: 1.72–5.19). Drug adverse effect (AHR = 6.1; 95% CI: 2.5, 14.34) and culture test result (AHR = 0.1; 95% CI: 0.1, 0.3) were independent predictors of LTF.

## Conclusion

This study concluded that drug-resistant tuberculosis mortality and LTF remains high in the study area. Continual support of the integration of TB/HIV service with emphasis and work to identified predictors may help in reducing drug-resistant tuberculosis mortality and LTF.

## Background

Drug resistant tuberculosis (DR-TB) is defined as resistance to any anti-TB drugs. Extensively drug-resistant tuberculosis (XDR-TB) is a subset of DR-TB with additional resistance to a fluoroquinolone and a second-line injectable agent [1].

According to the WHO, there were an estimated 484000 incident cases of DR/RR-TB in the year 2018 worldwide. Globally in 2018, an estimated 3.4% of new cases and 18% of previously treated cases had DR/RR-TB. A global total of 206,030 people with drug- or rifampicin-resistant TB (DR/RR-TB) were detected and notified in 2019, a 10% increase from 186,883 in 2018 [2,3].

The WHO estimates that about 5% of all TB cases progress to DR-TB, from which more than 40% died in 2013. Treating DR- TB and XDR-TB is particularly difficult even in resource-rich countries because the required long term treatment, a minimum of 18–24 months with second-line TB drugs, has significant adverse effects [4].

Drug resistant tuberculosis surveys in 2005 and 2014 and the 2018 WHO TB report in Ethiopia showed that the prevalence of DR-TB respectively, was 1.6%, 2.3%, and 2.7% among new and 11.8%, 17.8%, and 14% among previously treated TB cases [5–7].

Tuberculosis programmes have reported the LTF of two or more consecutive months to contribute to the poor levels of treatment success in DR-TB [5]. Although DR-TB treatment lasting for 20–24 months has shown to achieve a success rate of 50% of patients worldwide, recent advances show that a short course treatment regimen achieves high treatment success [3].

The 2020 WHO report [3] showed 0.71% and 12% respectively, as proportions of new cases and previously treated cases with DR/RR-TB in Ethiopia, making Ethiopia one among the countries with poorest treatment success rate when compared to other member states. Factors including LTF and poor adherence to treatment were reported to contributing to poor treatment outcome in some countries [3].

The Ethiopian Federal Ministry of Health adopted a two armed standardized regimen consisting of an 8-month intensive phase and a 12-month continuation phase. However, few observational studies have been conducted and reveal poor treatment outcomes including loss to follow-up among DR-TB patients [6]. Both patient- and regimen-related factors were associated with loss to follow-up, important information which may guide interventions to improve treatment adherence, particularly in the first 11 months [8]. Studies have also shown

that, similar to what happens with drug-susceptible TB patients, the LTF is one of the main problems in the treatment of DR-TB. In a study conducted in Brazil, identified number of determinants to loss to follow-up such as drug toxicity, long duration of treatment, and other social determinants [9].

Previous studies have shown that predictors of DR-TB mortality are: immune-suppression, tuberculosis related complications, malnutrition, HIV/AIDS, smoking cigarette, LTF, tuberculosis treatment history, and diabetes mellitus, for which, some are determined by patients' social-demographic, behavioural and lifestyle factors [10–12]. As far as is known to us, only one report has been published in Ethiopia to assess risks for mortality among DR-TB patients [13,14].

Limited studies conducted in Ethiopia showed lost to follow-up among DR-TB registered patients for treatment was high in the first 6 months compared to later follow-up months [12], and the incidence of mortality was 7.42/100 person-years follow-up during treatment and most patients died early on the initiation of treatment [15].

Therefore, the current study aimed to assess predictors, time to mortality and loss to follow-up among patients infected with drug resistance tuberculosis in Ethiopia.

## Methods

### Study design and study area

A retrospective follow up study was carried out from 01 November 2012 to 31 December 2017 among DR-TB patients in hospitals of Oromia regional state, Ethiopia where high prevalence of TB has been reported to be as high as 21.3%. Included hospitals were Metu Karl, Shanen Gibe, Bishoftu, Shashemene, Adama, Chiro, Deder and Nekemt. The 2016 WHO global report puts Ethiopia among countries with high burden of DR-TB, with more than 3,300 cases estimated annually. In 2016, it was estimated that 2.7% (1.5–4.0 of newly notified and 14% (3.6–25) of previously treated TB patients had DR-TB [16].

### Study population and inclusion criteria

The study population involved all DR-TB patients with complete chart, aged 18 years and above enrolled into the adult Tuberculosis clinics in the Oromia Hospitals from 01 November 2012 to 31 December 2017. The period was selected in order to have the nearest five year follow up study period and it was also the period when hospitals in Oromia started the full implementation of standardized formats, documentation and recording systems in a regular manner.

### Sample size determination

The sample size was calculated using single population proportion formula after considering the following assumptions: 50% proportion, 95% confidence level (Zα/2 = 1.96), 5% margin of error (d). The final sample size was 423 but the registered DR-TB patients at the study period was 497 and 91 patient charts were incomplete. Hence, we have included 406 patient charts.

### Data collection tools and procedures

Data were extracted by using a structured format that was developed from the standard treatment protocol for DR-TB treatment. The checklist sought from individual records, information on patient-related data (Socio demographic characteristics), clinical characteristic, comorbidities, treatment and others.

Selected nurses who worked in TB clinics extracted all DR-TB patients' data retrospectively (including date of death and LTF) from the hospitals' registration log books during January 1–30, 2018. Trained data collectors assessed the data quality using a pre-tested data collection tool, and two public health professionals provided continuous supervision and monitoring. Supervisors, data clerks and investigators checked the completeness and consistency of data before and after data entry.

## Data analysis

Data were entered into EPI data (version 4.6.0.0) and the analysis was conducted using Statistical package for Social Science (SPSS version 20.0) and Stata (version 14.0). Data were cleaned to remove irrelevant observations before analyses were performed. Median, mean and frequencies (as percentages) were used to describe patients' characteristics in each cohort. The response variable was survival time, defined as "time in months transpired from the date of initial DR-TB treatment to death/LTF" or, in the case of individuals who did not die/not LTF (censored), "the time in months transpired to LTF. The Kaplan-Meier curve was used to estimate the probability of death and LTF and the median time to death and LTF after the initiation of the DR-TB treatment. The log-rank test was used to compare time to death and LTF between the two groups. The Cox proportional hazard model was used to determine predictors of death and LTF after DR-TB diagnosis. All statistically significant (p < 0.05) variables in the multivariate proportional hazard cox regression model were considered as predictors for mortality and LTF. The crude and adjusted hazard ratio (HR) and its 95% confidence interval (CI) were estimated.

## Ethical issues

Ethical approval was obtained from the Review Ethics Committee of the School of Health Science at Madda Walabu University. The data were fully anonymized and name and identifications were not extracted. To preserve patient confidentiality, the identified patients' data were extracted from medical charts by nurses working in the respective TB clinic.

## Result

### Socio demographic characteristics

A total of 497 DR-TB patients were registered during the study period. Ninety one (91) patients were excluded due to missing charts, incomplete baseline and follow-up data. The median age of the included participants (406), the median age was 28 (IQR: 27.1, 29) years, and 147 (36.2%) individuals aged between 25–30 years old, made the largest group of participants. Two hundred and thirty nine (58.9%) were males and 41.1% were females. Among males, 41 died and 20 were LTF and individual aged 41 and above years old, recorded the highest death rate (Table 1).

### Clinical characteristics of the study participants

Almost all study participants had a diagnosis of pulmonary TBs 397 (97.8%), of which 71 died and 31 were LTF. About 2/3 of DR-TB patients had only rifampicin resistance 260 (64%), of whom 46 died and 23 were LTF. Almost all patients (99.8%), had treatment initiated after the confirmation of TB. Among HIV tested patents, 77 (19.1%) were positive for HIV of which 31 died and 4 were LTF. One tenth 41 (10.1%) of DR-TB patents had comorbidity, of whom 25 died and 7 were LTF. Forty nine (12.1%) DR-TB had unilateral lung cavity, of whom 12 died and 3 were LTF. About 11 (2.7%) patients had any type of addiction, of which 2.8% and 3.1%

**Table 1. Socio-demographic characteristics of drug resistance tuberculosis infected patents in Oromia region, Ethiopia, 2020.**

| Variables | | Number (406) | Mortality | | LTF | |
|---|---|---|---|---|---|---|
| | | | Event (n = 71) | Censored (n = 335) | Event (n = 32) | Censored(n = 374) |
| Gender | Male | 239 (58.9) | 41 | 198 | 20 | 219 |
| | Female | 167 (41.1) | 30 | 137 | 12 | 155 |
| Address | Urban | 261 (64.3) | 52 | 209 | 18 | 243 |
| | Rural | 145 (35.7) | 19 | 126 | 14 | 131 |
| Age | 19–24 | 117 (28.8) | 15 | 102 | 14 | 103 |
| | 25–30 | 147 (36.2) | 19 | 128 | 7 | 140 |
| | 31–40 | 79 (19.2) | 55 | 55 | 7 | 72 |
| | ≥41 | 63 (15.5) | 50 | 50 | 4 | 59 |

patients died and LTF respectively. At the beginning of treatment, the majority of patents had BMI $\leq$ 18 and after treatment a similar number of patients had BMI $\geq$ 18 (359 (88.4%) and 362 (89.2%) respectively) (Table 2).

## Magnitude and time to mortality and loss to follow up

Of the 406 study participants, 275 (67.7%) were cured, 71(17.5%) died, 32 (7.9%) LTF and the treatment failed in 13 (3.2%). (Fig 1) The incidence density of death in the cohort was 9.8 per 1000 person-months. The incidence density of loss to follow up in the cohort was 4.5 per 1000 person-months.

## Survival probability of DR-TB patients for mortality

A total of 406 DR-TB patients were followed for a minimum of 1 month and for a maximum of 24 months with mean follow up period of 21 months (CI: 20.4, 21.7) and contributed 7084 person months observations. The cumulative survival probability of patients at the end of 9 months was 87.4%, at the end of 19 months was 83.3%, and at the end of 24 months was 77.5% (Fig 2). For loss to follow-up, they had follow up for a minimum of one months and maximum of 23 months on follow up with mean follow up period of 22 months. The cumulative survival probability of patients at the end of 8 months was 96.5%, at the end of 19 months was 94.3%, and at the end of 24 months was 84.5% (Fig 3). Kaplan-Meier analysis revealed that LTF was higher in patients who were HIV-infected compared to non-infected (Fig 4). Additionally, patients with co infection had lower survival probability than their counterparts (Fig 5).

## Predictors of time to mortality and loss to follow up

According to multivariate Cox proportional hazard regression analysis model, two variables were found to be independent predictors for the mortality during the DR-TB patients' treatment, include chest radiographic finding and HIV sero status. The risk of death decreased by 63% in patients with normal chest radiographic finding compared to chest radiographic with massive effusion (AHR = 0.37, 95% CI: 0.17–0.79). With regards to HIV sero status, HIV positive DR-TB patients had three times higher risk of death compared to their counterpart, 2.98 (95% CI: 1.72–5.19) (Table 3). Patients who had drug adverse effect were six times at higher risk of LTF compared to patients who had no adverse drug effect (AHR = 6.1; 95% CI = 2.5, 14.34). Moreover, those who had negative culture result were 90% less likely to record LTF compared to patients had no culture done [AHR = 0.1; 95% CI: 0.1, 0.3] (Table 4).

**Table 2. Clinical characteristics of drug resistance tuberculosis infected patents in Oromia region, Ethiopia, 2020.**

| Variables | | Number (406) | Mortality | | LTF | |
|---|---|---|---|---|---|---|
| | | | Event (n = 71) | Censored (n = 335) | Event (n = 32) | Censored (n = 374) |
| Site of TB infection | Pulmonary | 397 (97.8%) | 71 | 326 | 31 | 366 |
| | Extra pulmonary | 9 (2.2%) | 0 | 9 | 1 | 8 |
| Type of resistance | RR-Rifampicin | 260 (64%) | 46 | 214 | 23 | 237 |
| | M-MDR | 143(35.2%) | 24 | 119 | 135 | 8 |
| | X-XDR | 3 (0.7%) | 1 | 2 | 2 | 1 |
| Smear result at zero month | Positive | 276 (68%) | 46 | 230 | 16 | 260 |
| | Negative | 62 (15.3%) | 6 | 56 | 10 | 52 |
| | Not done/result not available | 68 (16.7%) | 19 | 49 | 6 | 62 |
| Culture result at start | Positive | 252 (62.1%) | 34 | 218 | 19 | 233 |
| | Negative | 40 (9.9%) | 2 | 38 | 3 | 37 |
| | Not done/result not available | 114 (28.1%) | 35 | 79 | 10 | 104 |
| Culture result at the end of treatment | Positive | 18 (4.4%) | 7 | 11 | 3 | 15 |
| | Negative | 322 (79.3%) | 30 | 292 | 15 | 307 |
| | Not done/result not available | 66 (16.3%) | 34 | 32 | 14 | 52 |
| Smear result at the end of treatment | Positive | 14 (3.4%) | 7 | 7 | 2 | 12 |
| | Negative | 339(83%) | 35 | 304 | 20 | 319 |
| | Not done/result not available | 53 (13.1%) | 29 | 24 | 10 | 43 |
| Diagnostic method | Xpert/MTB/TB/RIF | 316 (77.8%) | 61 | 255 | 20 | 296 |
| | LPA | 68 (16.7%) | 9 | 59 | 8 | 60 |
| | Culture | 3 (0.7%) | 0 | 3 | 0 | 3 |
| | Others | 19 (4.7%) | 1 | 18 | 4 | 15 |
| Reason for entering MDR treatment | Bacteriological confirmed | 405 (99.8%) | 71 | 334 | 32 | 373 |
| | Clinically diagnosed | 1 (0.2%) | 0 | 1 | 0 | 1 |
| HIV tested | Yes | 402(99%) | 70 | 332 | 32 | 370 |
| | No | 2(4%) | 1 | 3 | 0 | 4 |
| HIV test result | Reactive | 77 (19.1%) | 31 | 46 | 4 | 73 |
| | None reactive | 326 (80.9%) | 39 | 287 | 28 | 298 |
| ART started | Yes | 30(98.8%) | 30 | 45 | 3 | 72 |
| | No | 1(3.2%) | 1 | 1 | 1 | 1 |
| Co-infections | Yes | 41 (10.1%) | 16 | 25 | 7 | 34 |
| | No | 365 (89.9%) | 55 | 310 | 25 | 340 |
| Chronic diseases | Yes | 12 (3%) | 4 | 8 | 2 | 10 |
| | No | 394 (97%) | 67 | 327 | 30 | 364 |
| Drug Adverse effect | Yes | 40 (9.9%) | 5 | 35 | 9 | 31 |
| | No | 366 (90.1%) | 66 | 300 | 23 | 343 |
| Treatment regimen | Standard | 401 (98.9%) | 70 | 331 | 31 | 370 |
| | Individualized | 5 (1.2%) | 1 | 4 | 1 | 4 |
| Steroid use | Yes | 8 (2%) | 2 | 6 | 0 | 8 |
| | No | 140 (34.5%) | 24 | 116 | 6 | 134 |
| | Unknown | 258 (63.5%) | 45 | 213 | 26 | 232 |
| Chest radiographic finding | Unilateral Cavity | 49 (12.1%) | 12 | 37 | 3 | 46 |
| | Bilateral Cavity | 28 (6.9%) | 9 | 19 | 4 | 24 |
| | Abnormality without cavity | 128 (31.5%) | 11 | 117 | 7 | 121 |
| | Massive effusion | 24 (5.9%) | 6 | 18 | 3 | 21 |
| | Others | 177 (43.6%) | 33 | 144 | 15 | 162 |

(*Continued*)

**Table 2.** (Continued)

| Variables | | Number (406) | Mortality | | LTF | |
|---|---|---|---|---|---|---|
| | | | Event (n = 71) | Censored (n = 335) | Event (n = 32) | Censored (n = 374) |
| Any type of addiction | Yes | 11 (2.7%) | 2 | 9 | 1 | 10 |
| | No | 395 (97.3%) | 69 | 326 | 31 | 364 |
| BMI at start of treatment | 18 | 359 (88.4%) | 67 | 292 | 30 | 329 |
| | >18 | 47 (11.6%) | 4 | 43 | 2 | 45 |
| BMI at end of treatment | = <18 | 44 (10.8%) | 10 | 34 | 4 | 40 |
| | >18 | 362 (89.2%) | 61 | 301 | 28 | 334 |

## Discussion

This study was designed to assess the incidence rate and to identify the predictors of mortality and lost to follow-up among DR-TB patients attending their treatment at the Oromia Hospitals, Ethiopia. A total of 406 DR-TB patients were followed and produced 7084 person-months observations and 71 (17.5%) died. This finding was higher than a study conducted in Dangila and agreed with a systematic review and meta-analysis conducted in Ethiopia [17,18].

The patients who were LTF would have not completed their treatment regime, resulting in serious public health problems because these patients are at higher risk of extra DR-TB and transmission to the community [19]. In this study, 32 (7.9%) for DR-TB treatment enrolled patents were LTF. Similar high rate of LTF among DR-TB patients has been demonstrated in a systematic review and other studies conducted in China, Pakistan, and Brazil [20–23], together illustrating the immense challenges in achieving completion of currently recommended treatment regimens for DR-TB. In this study LTF rate was also substantially higher than the WHO recommended target of 5% [24], which implies the need to demand for more comprehensive

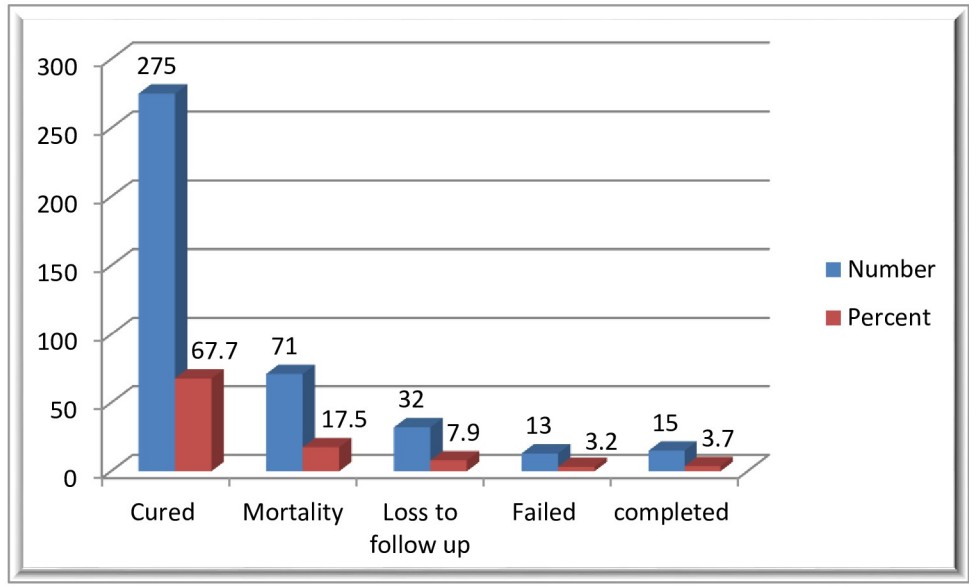

**Fig 1. Treatment outcomes of drug resistance tuberculosis infected patents in Oromia region, Ethiopia, 2020.**

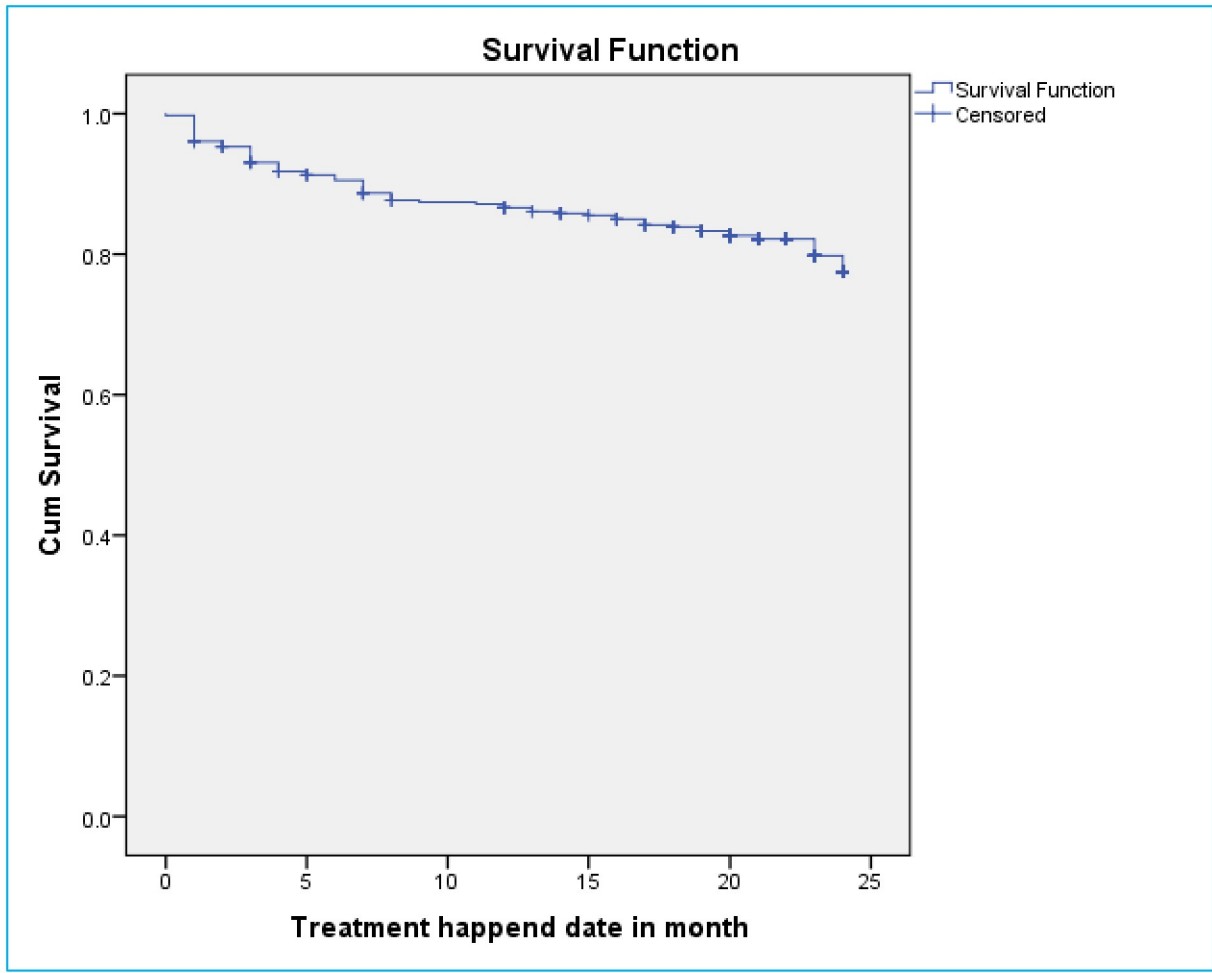

**Fig 2. Kaplan–Meier survival estimate for mortality of drug resistance tuberculosis infected patents in Oromia region, Ethiopia, 2020.**

approaches to reduce LTF among DR-TB patients by targeting those risk factors for treatment interruptions.

The incidence density of LTF in the current study cohort was 4.5 per 1000 person-months. This finding was lower than a study conducted in Northern part of Ethiopia and higher than a study conducted in St. Peter's specialized Tuberculosis treatment hospital in Ethiopia [13,25]. We hypothesise that this variation could possibly be due to differences, including in the sample size, study population, treatment guideline, and regimens.

The incidence density of death in the cohort was 9.8 per 1000 person-months. This finding was higher than a study conducted in Amhara region as well as a study conducted in nation-wide in Ethiopia [12,26]. The possible explanation for this might be due to the difference in the follow-up periods, because the longer follow up periods have the probability of decreasing the occurrence of events.

The risk of death decreased by 63% in patients with normal chest radiographic findings compered to chest radiographic finding with massive effusion. This finding was consistent with previous studies conducted in Tanzania and central Ethiopia [15,27]. The lower respiratory tract infections are known to have consequences including the accumulation of

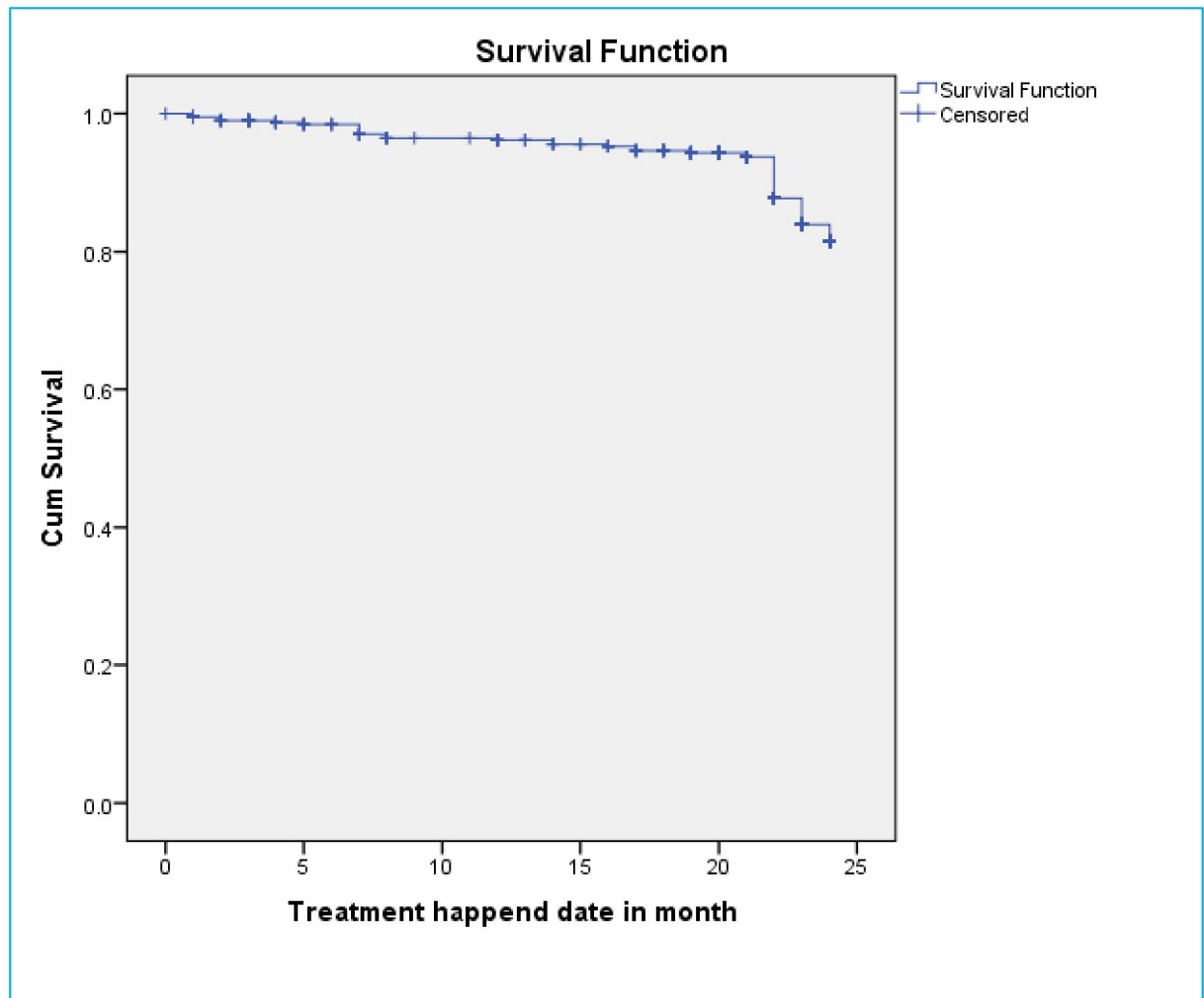

**Fig 3. Kaplan–Meier survival estimate for loss to follow up of drug resistance tuberculosis infected patents in Oromia region, Ethiopia, 2020.**

inflammatory exudate in the alveoli, resulting in reduced oxygen exchange and respiratory insufficiency, and the DR-TB patients with clinical complications experience longer recovery times and poor response to anti TB-medications [28,29].

In this study, HIV positive DR-TB patients had 2.98 times higher risk of death compared to the HIV negative DR-TB, findings which are consistent with studies conducted in other sites including: Amhara region, Dangila, St. Peter's specialized Tuberculosis hospital in Ethiopia, eastern and central Africa, and in Ethiopia, Eastern Europe, Brazil, Peru, Nigeria [12,17,25,26,30–34]. The high mortality rate of DR-TB patients among HIV positive partici-pants might be due to the synergistic effects of the two co-infections, i.e. HIV and DR-TB. Because DR-TB patients receive treatment for a minimum of two years, they might experience serious drug adverse effects and toxicities due to the high burden of pills, which may result in poor adherence to treatment and poorer overall patients' poor outcome. Coupled with the known higher rate of smear negative in HIV positive patients, the diagnosis of DR-TB in HIV positive patients is said to difficult as may be confused with other pulmonary or systemic

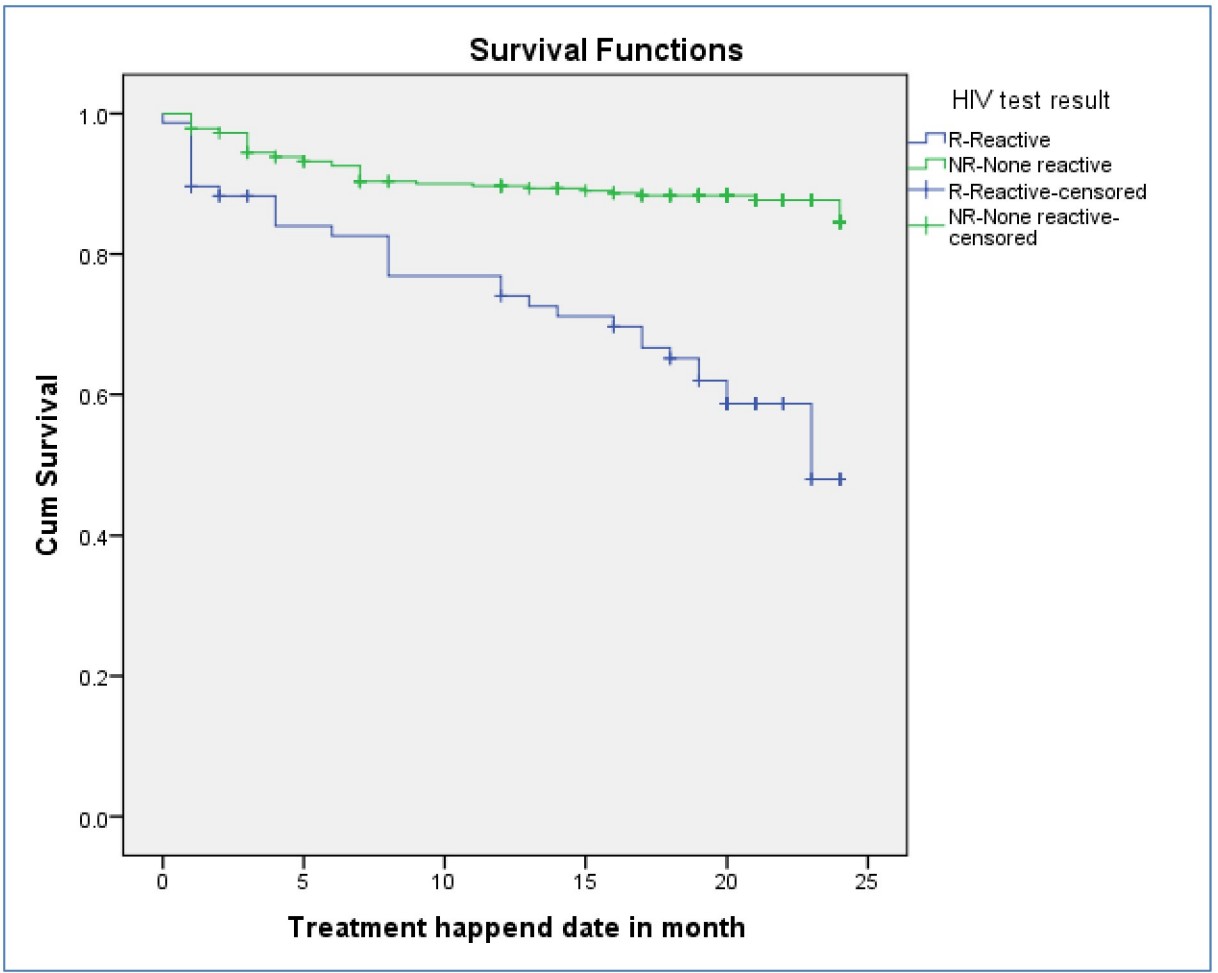

**Fig 4. Kaplan–Meier survival estimate for LTF of drug resistance tuberculosis infected patents for HIV reactive and non-reactive patients in Oromia region, Ethiopia, 2020.**

infections. These interactions, can result in misdiagnosis or delays in a diagnosis and leading to higher morbidity and mortality [35,36].

Patients who had drug adverse effect were six times higher risk of LTF compared to patients who had no adverse drug effect. This finding was consistent with studies that were conducted in the Philippines, Pakistan, Tajikistan, Georgia, and a systematic review and Meta-analysis [8,21,37–39]. These findings support the knowledge that an association exists between LTF and individuals taking DR-TB drugs since second line drugs are more toxic and more likely to cause severe side effects.

Drug resistance Tuberculosis patients who had negative culture result at the end of initiation phase were 90% less likely to be LTF compared to patients who had no culture undertaken. This seems to be promising result as at the end of initiation phase could motivate patients to finish treatment, and thus a possibility of positive outcome.

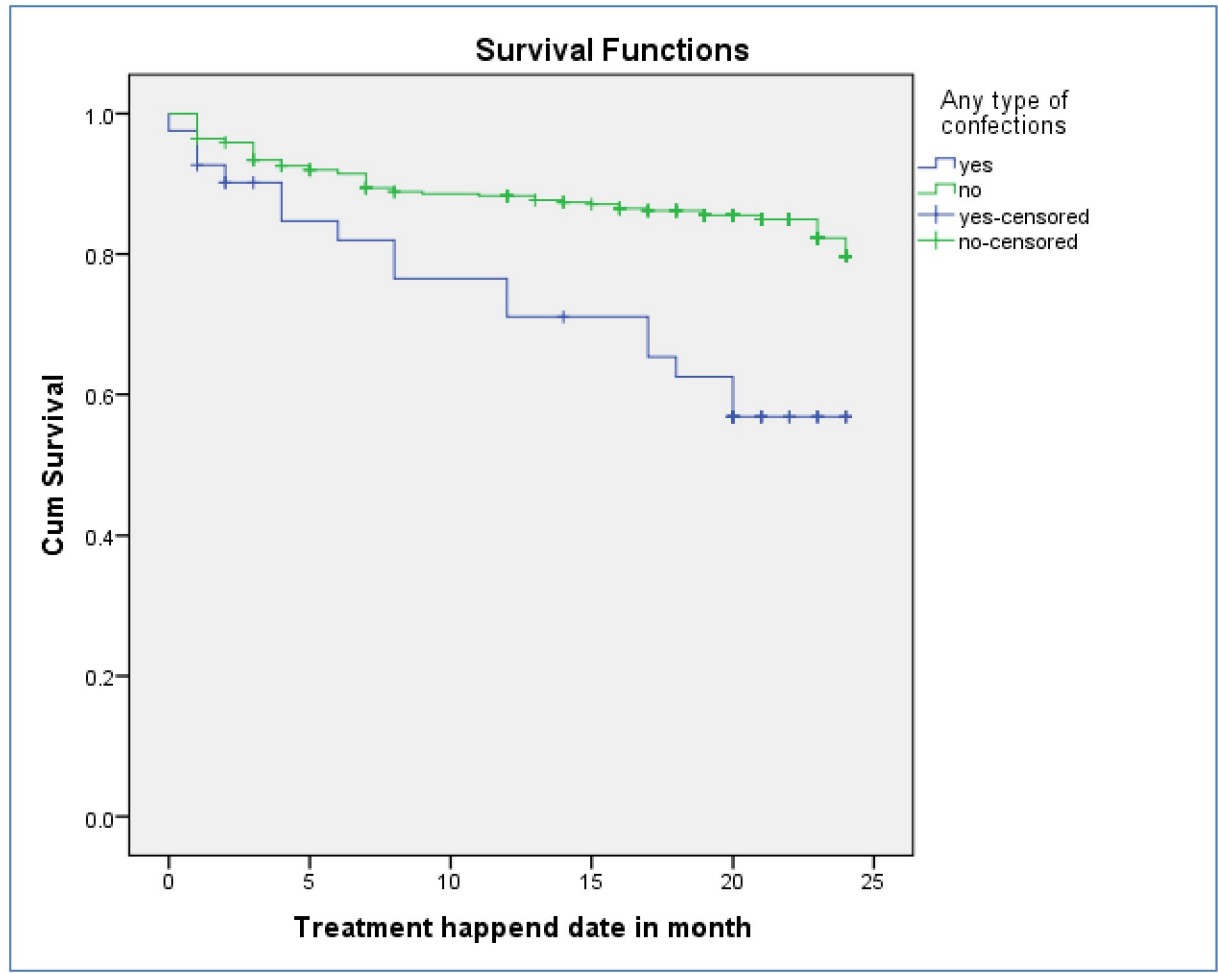

**Fig 5. Kaplan–Meier survival estimate for mortality of drug resistance tuberculosis infected patents for patients with co morbidity and without co morbidity in Oromia region, Ethiopia, 2020.**

## Limitations of the study

As a limitation, this study did not considered some predictors such as viral load, liver function and renal function tests because it was conducted using secondary data that have missing records. Additionally, there could also be underestimation of the mortality incidence rate and LTF due to record problems.

## Conclusion and recommendation

Our findings point to some risk factors related to mortality and loss to follow-up, which are public health problems that contribute to worsening DR-TB treatment. It is important for TB control programs and health professionals to identify predictors in patients with increased risk of loss to follow-up and to adopt specific strategies to address this problem and prevent death. Continual support of the integration of TB/HIV service with emphasis and further work on identified predictors may help in reducing DR-TB mortality and LTF.

**Table 3. Predictors of mortality among drug resistance tuberculosis infected patents in Oromia region, Ethiopia, 2020.**

| Variables | | No. at risk | No. of death | CHR (95% CI) | AHR (95% CI) |
|---|---|---|---|---|---|
| Address | Urban | 261 | 52 | 1.5(0.1, 2.6) | 1.06(0.62, 1.83) |
| | Rural | 145 | 19 | 1 | 1 |
| HIV sero status | **Positive** | 2 | 1 | 3.9(2.4, 6.2) | **2.98(1.72, 5.19)**\* |
| | Negative | 402 | 70 | 1 | 1 |
| Confection | Yes | 41 | 16 | 2.9(1.7, 5.2) | 1.616(0.82, 3.20) |
| | No | 365 | 55 | 1 | 1 |
| Chronic diseases | Yes | 12 | 4 | 1.9 (0.7, 5.3) | 2.57(0.88, 7.51) |
| | No | 394 | 67 | 1 | 1 |
| Drug adverse effect | Yes | 40 | 5 | 0.7(0.3, 1.7) | 0.65(0.26, 1.65) |
| | No | 366 | 66 | 1 | 1 |
| Chest radiographic finding | Massive effusion | 49 | 12 | 1 | 1 |
| | Bilateral Cavity | 28 | 9 | 1.3(0.7, 2.5) | 1.60(0.79, 3.24) |
| | Abnormality without cavity | 128 | 11 | 1.9(0.9, 3.9) | 2.09(0.95, 4.66) |
| | Normal | 24 | 6 | 0.4(0.2, 0.8) | **0.37(0.17, 0.79)**\* |
| | Unilateral Cavity | 177 | 33 | 1.3(0.6, 3.2) | 1.10 (0.44, 2.76) |
| BMI at start of treatment | = <18 | 44 | 10 | 2.5(0.9, 6.9) | 1.7(0.83, 3.51) |
| | >18 | 362 | 61 | 1 | 1 |
| BMI at end of treatment | = <18 | 359 | 67 | 1.3(0.7, 2.6) | 1.88(0.65, 5.49) |
| | >18 | 47 | 4 | 1 | 1 |

Notes:

\*Significant at p<0.05.

**Table 4. Predictors of LTF among drug resistance tuberculosis infected patents in Oromia region, Ethiopia, 2020.**

| Variable | | No. at risk | No. of LTF | CHR (95% CI) | AHR (95% CI) |
|---|---|---|---|---|---|
| Address | Urban | 261 | 18 | 0.6 (0.3, 1.2) | 0.5 (0.3, 1.2) |
| | Rural | 145 | 14 | 1 | 1 |
| Confection | Yes | 41 | 7 | 3.1(1.3, 7.1) | 1.6 (0.6, 3.8) |
| | No | 365 | 25 | 1 | 1 |
| Drug A/E | Yes | 40 | 9 | 4.1(1.9, 8.8) | **6.1 (2.5, 14.34)**\* |
| | No | 366 | 23 | 1 | 1 |
| Steroid use | Yes | 8 | 0 | | |
| | No | 140 | 6 | | |
| | Unknown | 258 | 26 | | |
| BMI at start | = <18 | 44 | 30 | 2.8(0.7, 11.8) | 1.5 (0.3, 6.7) |
| | >18 | 362 | 2 | 1 | 1 |
| Sex | Male | 239 | 20 | 1.2(0.6, 2.5) | 1.5 (0.6, 3.2) |
| | Female | 167 | 12 | 1 | 1 |
| Culture at end | Positive | 18 | 3 | 0.6(0.2, 1.9) | 1.0 (0.3, 3.9) |
| | Negative | 322 | 15 | 0.13(0.06, 0.28) | **0.1 (0.1, 0.3)** |
| | Not done/result not available | 66 | 14 | 1 | 1 |

Notes:

\*Significant at p<0.05.

## Supporting information

**S1 Data. Available data for predictors of mortality and loss to follow-up among drug resistant tuberculosis patients in Oromia Hospitals, Ethiopia.**
(SAV)

## Author Contributions

**Conceptualization:** Demelash Woldeyohannes.

**Data curation:** Demelash Woldeyohannes, Yohannes Tekalegn.

**Formal analysis:** Demelash Woldeyohannes.

**Funding acquisition:** Demelash Woldeyohannes.

**Investigation:** Demelash Woldeyohannes.

**Methodology:** Demelash Woldeyohannes.

**Project administration:** Demelash Woldeyohannes.

**Resources:** Demelash Woldeyohannes.

**Software:** Demelash Woldeyohannes.

**Supervision:** Demelash Woldeyohannes.

**Validation:** Demelash Woldeyohannes.

**Visualization:** Demelash Woldeyohannes.

**Writing – original draft:** Demelash Woldeyohannes, Zeleke Hailemariam.

**Writing – review & editing:** Demelash Woldeyohannes, Yohannes Tekalegn, Biniyam Sahiledengle, Tesfaye Assefa, Rameto Aman, Zeleke Hailemariam, Lillian Mwanri, Alemu Girma.

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
