## [Decision Letter · Decision Letter 0]

17 Mar 2021

PONE-D-21-04578

Predictors of Mortality and Loss to Follow-up among Drug Resistant Tuberculosis Patients in Ethiopia: A Retrospective Follow-up Study

PLOS ONE

Dear Dr. Demelash Woldeyohannes,

Thank you for submitting your manuscript to PLOS ONE. After careful consideration, we feel that it has merit but does not fully meet PLOS ONE’s publication criteria as it currently stands. Therefore, we invite you to submit a revised version of the manuscript that addresses the points raised during the review process.

We look forward to receiving your revised manuscript.

Kind regards,

Claudia Marotta

Academic Editor

PLOS ONE

Journal Requirements:

We will update your Data Availability statement on your behalf to reflect the information you provide-

4. In the ethics statement in the manuscript and in the online submission form, please provide additional information about the patient records used in your retrospective study, including: a) whether all data were fully anonymized before you accessed them; b) the date range (month and year) during which patients' medical records were accessed; c) the date range (month and year) during which patients whose medical records were selected for this study sought treatment. If the ethics committee waived the need for informed consent, or patients provided informed written consent to have data from their medical records used in research, please include this information.

5. You indicated that you had ethical approval for your study. In your Methods section, please ensure you have also stated whether you obtained consent from parents or guardians of the minors included in the study or whether the research ethics committee or IRB specifically waived the need for their consent.

6.Thank you for submitting the above manuscript to PLOS ONE. During our internal evaluation of the manuscript, we found some minor occurrences of overlapping text with the following previous publication(s), some of which you are an author, which needs to be addressed:

- https://doi.org/10.1590/0102-311X00048217

- https://doi.org/10.1186/1471-2334-13-297

We would like to make you aware that copying extracts from previous publications word-for-word, especially outside the methods section, is unacceptable. In addition, the reproduction of text from published reports has implications for the copyright that may apply to the publications.

Please revise the manuscript to quote or rephrase the duplicated text and cite your sources for text outside the methods section. Please note that further consideration is dependent on the submission of a manuscript that addresses these concerns about the overlap in text with published work.

Additional Editor Comments:

dear Auhtors follow reviewer suggestions to improve your paper

Reviewers' comments:

Reviewer's Responses to Questions

**Comments to the Author**

1. Is the manuscript technically sound, and do the data support the conclusions?

Reviewer #1: Yes

Reviewer #2: Yes

2. Has the statistical analysis been performed appropriately and rigorously? 

Reviewer #1: Yes

Reviewer #2: Yes

3. Have the authors made all data underlying the findings in their manuscript fully available?

Reviewer #1: Yes

Reviewer #2: Yes

4. Is the manuscript presented in an intelligible fashion and written in standard English?

Reviewer #1: Yes

Reviewer #2: Yes

5. Review Comments to the Author

Reviewer #1: This paper evaluated the incidence and predictors of mortality and lost to follow-up among Rifampin resistant TB patients in a state of Ethiopia. RR/MDR-TB is a major public health problem all over the world. The authors presented us the severity and poor outcome of RR/MDR-TB in Oromia regional state of Ethiopia which is common in many high burden countries. The article could help the readers to update the factors for the treatment outcome in this country. However, there some questions need to be addressed for this paper.

1. The definition for DR-TB was not correctly described in the background. Drug resistant tuberculosis (DR-TB) was referred to resistance to any anti-TB drugs, and resistance to at least Isoniazid (H) and rifampin(R) was defined as MDR-TB according to WHO. The definition needs to be clear for the entire study.

2. The study was only conducted in in a state of Ethiopia, thus cannot represent the whole country of Ethiopia. The title and conclusion of the article should be more precisely set and described.

3. Although the study had been approved by the ethics committee, it had used the patient retrospective data which the patient consent exempt is also needed for the study. Was it also approved by the committee?

4. According to the study, 91 patients were excluded due to missing charts, incomplete baseline and follow-up data. How many patients among these excluded group died or LTF? Because the poor outcome could also cause the missing of data.

5. Page 8, line 5, there was a mistake of “of which 7 died…”. It should be 71 died.

Reviewer #2: I read with great interest this paper.Demelash Woldeyohannes and colleague wrote on important global health problem: Predictors of Mortality and Loss to Follow-up among Drug Resistant Tuberculosis in high burden setting.

Below my suggestions:

1. Introduction: well that article show data on TB global burden data of TB Report 2020. Add the role of social determinants of health in worst outcome of TB, therapy failure and onset on TB MDR (see and cite Social determinants of therapy failure and multi drug resistance among people with tuberculosis: A review. Tuberculosis (Edinb). 2017 Mar;103:44-51. doi: 10.1016/j.tube.2017.01.002.2. Methods: clear. Correct tuberculosis in TB or Tuberculosis.

3. Results: no suggestions

4. Discussion: discuss better the role of age (Active Pulmonary Tuberculosis in Elderly Patients: A 2016-2019 Retrospective Analysis from an Italian Referral Hospital. Antibiotics (Basel) in worse outcome and onset adverse events and how they need more medical attention and the role of comorbidity exspecially diabetes (Diabetes in active tuberculosis in low-income countries: to test or to take care? Lancet Glob Health. 2019 Jun;7(6):e707. doi: 10.1016/S2214-109X(19)30173-1. PMID: 31097272.). In fact, Many factors contribute to the lack of efficacy in contain- ing TB, as social determinants of health (SDH) and con- comitance of other co-morbidities, such as co-infections (in particular with HIV) and non-communicable diseases, such as diabetes mellitus (DM). Furthermore, TB and DM remain serious public health problems particularly in low‐ and middle‐income countries. The coexistence of these two conditions is an example of a bidirectional association between a communicable and non‐communicable disease that increases the dual burden of DM and TB(see and cite Pizzol D, et al. Prevalence of diabetes mellitus in newly diagnosed pulmonary tuberculosis in Beira, Mozambique. Afr Health Sci. 2017 Sep;17(3):773-779. )

Compare better your data with other data from low income country anger high burden of Tb (es Mozambique) (Predictors of therapy failure in newly diagnosed pulmonary tuberculosis cases in Beira, Mozambique. BMC Res Notes. 2018 Feb 5;11(1):99. doi: 10.1186/s13104-018-3209-9. PMID: 29402317; PMCID: PMC5800087)

6. PLOS authors have the option to publish the peer review history of their article (what does this mean?). If published, this will include your full peer review and any attached files.

Reviewer #1: No

Reviewer #2: **Yes: **Francesco Di Gennaro

---

## [Author Response · Author response to Decision Letter 0]

12 Apr 2021

Responses for the comments from reviewers

We are very grateful to the editor and reviewers for their valuable comments and we believe that the revised manuscript is much stronger as a result of their feedback.

Dear Editor and reviewers:

Thank you for the opportunity to revise our manuscript “Predictors of Mortality and Loss to Follow-up among Drug Resistant Tuberculosis Patients in Ethiopia: A Retrospective Follow-up Study” (PONE-D-21-04578) for publication by PLOS ONE.

We have addressed the Editors’ and reviewers’ comments in our responses and incorporated related changes into the manuscript. For convenience, the reviewers’ comments are in black and our responses are in blue. 

The primary changes made to the study include: Editing of the manuscript to improve clarity and consistency

Additional requirements

1. Please ensure that your manuscript meets PLOS ONE's style requirements

Response: Thank you very much for reminding us to stick on the guideline, the revised manuscript meets PLOS ONE's style requirements 

2. Regarding data availability

Response: Thank you very much for flagging this point, all data are fully available without restriction 

3. Regarding financial disclosure 

Response: “The authors received no specific funding for this work.”

4. Regarding ethics statement 

Response: We have addressed the revised point regarding the ethics statement in the revised manuscript and cover letter. 

5. Please ensure you have also stated whether you obtained consent from parents or guardians of the minors included in the study or whether the research ethics committee or IRB specifically waived the need for their consent.

Response: This study didn’t included the minors 

6. Minor occurrences of overlapping text with the following previous publication: 

Response: We are grateful to the editor comments; we have paraphrased the overlapping statements 

Reviewer #1: 

1.1. The definition for DR-TB was not correctly described in the background. Drug resistant tuberculosis (DR-TB) was referred to resistance to any anti-TB drugs, and resistance to at least Isoniazid (H) and rifampin(R) was defined as MDR-TB according to WHO. The definition needs to be clear for the entire study.

Response: We are grateful to the reviewer comments; we made a correction in background of the study

1.2. The study was only conducted in in a state of Ethiopia, thus cannot represent the whole country of Ethiopia. The title and conclusion of the article should be more precisely set and described.

Response: We thank the reviewer for catching this. We made correction on the title of the study

1.3. Although the study had been approved by the ethics committee, it had used the patient retrospective data which the patient consent exempt is also needed for the study. Was it also approved by the committee?

Response: Yes, the ethical approval was received after the considering these issues 

1.4. According to the study, 91 patients were excluded due to missing charts, incomplete baseline and follow-up data. How many patients among these excluded group died or LTF? Because the poor outcome could also cause the missing of data.

Response: Thank you very much, 91 patent (missing) charts not included outcomes

1.5. Page 8, line 5, there was a mistake of “of which 7 died…”. It should be 71 died.

Response: Thank you for flagging this, we effected in the manuscript (Result section, line 5, page 8)

Reviewer #2

2.1. Introduction: well that article show data on TB global burden data of TB Report 2020. Add the role of social determinants of health in worst outcome of TB, therapy failure and onset on TB MDR (see and cite Social determinants of therapy failure and multi drug resistance among people with tuberculosis: A review. Tuberculosis (Edinb). 2017 Mar;103:44-51. doi: 10.1016/j.tube.2017.01.002.2. 

Response: We are grateful to the Reviewer for this comments, we have considered the factors and cited manuscript

2.2. Methods: clear. Correct tuberculosis in TB or Tuberculosis.

Response: We thank the reviewer for catching this. We made correction in the whole document

2.3. Results: no suggestions

2.4. Discussion: discuss better the role of age (Active Pulmonary Tuberculosis in Elderly Patients: A 2016-2019 Retrospective Analysis from an Italian Referral Hospital. Antibiotics (Basel) in worse outcome and onset adverse events and how they need more medical attention and the role of comorbidity exspecially diabetes (Diabetes in active tuberculosis in low-income countries: to test or to take care? Lancet Glob Health. 2019 Jun;7(6):e707. doi: 10.1016/S2214-109X(19)30173-1. PMID: 31097272.). In fact, Many factors contribute to the lack of efficacy in contain- ing TB, as social determinants of health (SDH) and con- comitance of other co-morbidities, such as co-infections (in particular with HIV) and non-communicable diseases, such as diabetes mellitus (DM). Furthermore, TB and DM remain serious public health problems particularly in low‐ and middle‐income countries. The coexistence of these two conditions is an example of a bidirectional association between a communicable and non‐communicable disease that increases the dual burden of DM and TB (see and cite Pizzol D, et al. Prevalence of diabetes mellitus in newly diagnosed pulmonary tuberculosis in Beira, Mozambique. Afr Health Sci. 2017 Sep;17(3):773-779.)

Response: We are grateful to the reviewer comments; our study included co infection such as HIV but it didn’t included non-communicable disease specifically diabetes mellitus and role of age. We take these concerns as the limitations of the study. 

2.5. Compare better your data with other data from low income country anger high burden of Tb (es Mozambique) (Predictors of therapy failure in newly diagnosed pulmonary tuberculosis cases in Beira, Mozambique. BMC Res Notes. 2018 Feb 5;11(1):99. doi: 10.1186/s13104-018-3209-9. PMID: 29402317; PMCID: PMC5800087)

Response: Thank you very much for flagging this. The outcome variable for the study conducted in Zimbabwe was therapy failure our study outcome variable were mortality and Loss to follow up, hence we unable to compare and contrast with our study findings. 

Please contact me again if additional information and editing required

Kind regards, 

Demelash Woldeyohannes

The second Responses for the comments from editor and reviewers

We are very grateful to the editor and reviewers for their valuable comments and we believe that the revised manuscript is much stronger as a result of their feedback.

Dear Editor and reviewers:

Thank you for the opportunity to revise our manuscript “Predictors of Mortality and Loss to Follow-up among Drug Resistant Tuberculosis Patients in Oromia Hospitals, Ethiopia: A Retrospective Follow-up Study” (PONE-D-21-04578R1) for publication by PLOS ONE.

We have addressed the Editors’ and reviewers’ comments in our responses and incorporated related changes into the manuscript. For convenience, the reviewers’ comments are in black and our responses are in blue. 

The primary changes made to the study include: Editing of the manuscript to improve clarity and consistency

1. According to Table 1, the study included data from minors (age <18). In your Methods section, please ensure you have also stated whether you obtained consent from parents or guardians of the minors included in the study or whether the research ethics committee or IRB specifically waived the need for parental consent.

Responses: Thank very much for flagging this: Under age 18 study participants were included due to editorial problem. They are under age category of 19 to 24. 

2. Please amend the title either on the online submission form or in your manuscript so that they are identical.

Responses: We made correction

3. Please ensure that you refer to Figure 2 and Figure 3 in your text as, if accepted, production will need this reference to link the reader to the figures.

Responses: We made correction

4. Please upload a copy of Supporting Information Table S1 - S4 which you refer to in your text.

Responses: We made correction

5. Thank you for updating your data availability statement. You note that your data are available within the Supporting Information files, but no such files have been included with your submission. At this time we ask that you please upload your minimal data set as a Supporting Information file, or to a public repository such as Figshare or Dryad. Please also ensure that when you upload your file you include separate captions for your supplementary files at the end of your manuscript.

Responses: We have uploaded as supplementary file

---

## [Decision Letter · Decision Letter 1]

14 Apr 2021

Predictors of Mortality and Loss to Follow-up among Drug Resistant Tuberculosis Patients in Oromia Hospitals, Ethiopia: A Retrospective Follow-up Study

PONE-D-21-04578R1

Dear Dr. Demelash Woldeyohannes,

We’re pleased to inform you that your manuscript has been judged scientifically suitable for publication and will be formally accepted for publication once it meets all outstanding technical requirements.

Kind regards,

Claudia Marotta

Academic Editor

PLOS ONE

Additional Editor Comments (optional):

congratulations

Reviewers' comments:

Reviewer's Responses to Questions

**Comments to the Author**

1. If the authors have adequately addressed your comments raised in a previous round of review and you feel that this manuscript is now acceptable for publication, you may indicate that here to bypass the “Comments to the Author” section, enter your conflict of interest statement in the “Confidential to Editor” section, and submit your "Accept" recommendation.

Reviewer #1: All comments have been addressed

Reviewer #2: All comments have been addressed

2. Is the manuscript technically sound, and do the data support the conclusions?

Reviewer #1: Yes

Reviewer #2: Yes

3. Has the statistical analysis been performed appropriately and rigorously? 

Reviewer #1: Yes

Reviewer #2: Yes

4. Have the authors made all data underlying the findings in their manuscript fully available?

Reviewer #1: Yes

Reviewer #2: Yes

5. Is the manuscript presented in an intelligible fashion and written in standard English?

Reviewer #1: Yes

Reviewer #2: Yes

6. Review Comments to the Author

Reviewer #1: (No Response)

Reviewer #2: (No Response)

7. PLOS authors have the option to publish the peer review history of their article (what does this mean?). If published, this will include your full peer review and any attached files.

Reviewer #1: No

Reviewer #2: No

---

## [Editor Report · Acceptance letter]

19 Apr 2021

PONE-D-21-04578R1 

Predictors of Mortality and Loss to Follow-up among Drug Resistant Tuberculosis Patients in Oromia Hospitals, Ethiopia: A Retrospective Follow-up Study 

Dear Dr. Woldeyohannes:

I'm pleased to inform you that your manuscript has been deemed suitable for publication in PLOS ONE. Congratulations! Your manuscript is now with our production department. 

Kind regards, 

on behalf of

Dr. Claudia Marotta 

Academic Editor

PLOS ONE